# Eurasian aspen (*Populus tremula* L.): Central Europe's keystone species 'hiding in plain sight'

**Antonín Kusbach[1]\*, Jan Šebesta[1], Robert Hruban[2], Pavel Peška[1,2], Paul C. Rogers[3]**

**1** Department of Forest Botany, Dendrology and Geobiocoenology, Faculty of Forestry and Wood Technology, Mendel University in Brno, Brno, Czech Republic, **2** Forest Management Institute, Brandýs nad Labem, Kroměříž, Czech Republic, **3** Western Aspen Alliance, Department of Environment & Society, and Ecology Center, Utah State University, Logan, Utah, United States of America

\* kusbach@mendelu.cz

**Data Availability Statement:** All relevant data are within the manuscript and its Supporting information files.

**Funding:** The author(s) received no specific funding for this work.

## Abstract

Knowledge of Eurasian aspen's (*Populus tremula* L.) ecological and growth characteristics is of high importance to plant and wildlife community ecology, and noncommercial forest ecosystem services. This research assessed these characteristics, identified aspen's habitat optimum, and examined causality of its current scarce distribution in central Europe. We analyzed a robust database of field measurements (4,656,130 stands) for forest management planning over 78,000 km$^2$ of the Czech territory. Our analysis we used GIS techniques, with basic and multivariate statistics such as general linear models, ordination, and classification. Results describe a species of broad ecological amplitude that has heretofore attracted little research attention. Spatial analysis showed significant differences between aspen and other forest non-forest cover types. Additionally, we found significant association between the proportion of aspen in a stand, the size of forest property, and the forest category. The results demonstrate historic reasons for aspen's widespread presence, though contemporary occurrence is limited. This study advances the concept of a quantitatively based aspen ecological optimum (niche), which we believe may be beneficial for numerous aspen associates in the context of anticipated warming. Irrespective of local ecology (i.e., the realized aspen niche), the study confirms that profit-driven policy in forestry is chiefly responsible for historic aspen denudation in central Europe. Even so, we demonstrate that ample habitat is present. Further solutions for improving aspen resilience are provided to support these keystone systems so vital to myriad dependent flora and fauna.

## Introduction

A recent massive decline of Norway spruce (*Picea abies*) and Scotch pine (*Pinus sylvestris*) forests has been generating huge treeless areas in central Europe, disrupting ecological bonds, landscape vistas, and causing real socio-economic issues [1]. Forest management looks for efficient ways to reforest large clear cuts and ensure the post-treatment ecological stability. Forest

**Competing interests:** The authors have declared that no competing interests exist.

ecologists refer to aspen (*Populus tremula* L.) (Fig 1) as an introductory, pioneer, or seral tree species because it prepares a site for subsequent forest types [2–4]. The use of resilient seral species, which resist climatic extremes, enhance site environment, and are a coherent part of natural forests dynamics, seems to be a potential solution for landscape rehabilitation. Besides other generalists such as birch, alder, or mountain ash, Eurasian or common aspen, the hardiest one [5], is not frequent in central Europe [6]. Aspen is generally absent due to historically low interest and conventionally accepted pioneer (seral) status, disqualifying the species from commercial interests for more than two centuries. The common practice of monotypic conifer management has left research and management gaps for aspen [7] and many seral species. After a genetic boom oriented toward productivity of aspen hybrids in 1970s and 80s [8], there has been limited research focused on this species. For example, in the Czech Republic (CZ), the most recent previous article on aspen was published 33 years ago [9]. Even in broad, worldwide reviews, besides commercially interesting species and fast-growing birches, poplars, and willows, there has been little attention devoted to European aspen [10]. The importance of aspen has been overlooked for a long time and both research and management concentrations have languished [9].

Eurasian aspen is a deciduous tree ranging from western and northern Europe to the Far East. This species is the most widespread tree species globally [5, 11]. It comprises a range of subspecies and ecotypes; these populations grow in Iceland, the United Kingdom, Scandinavia,

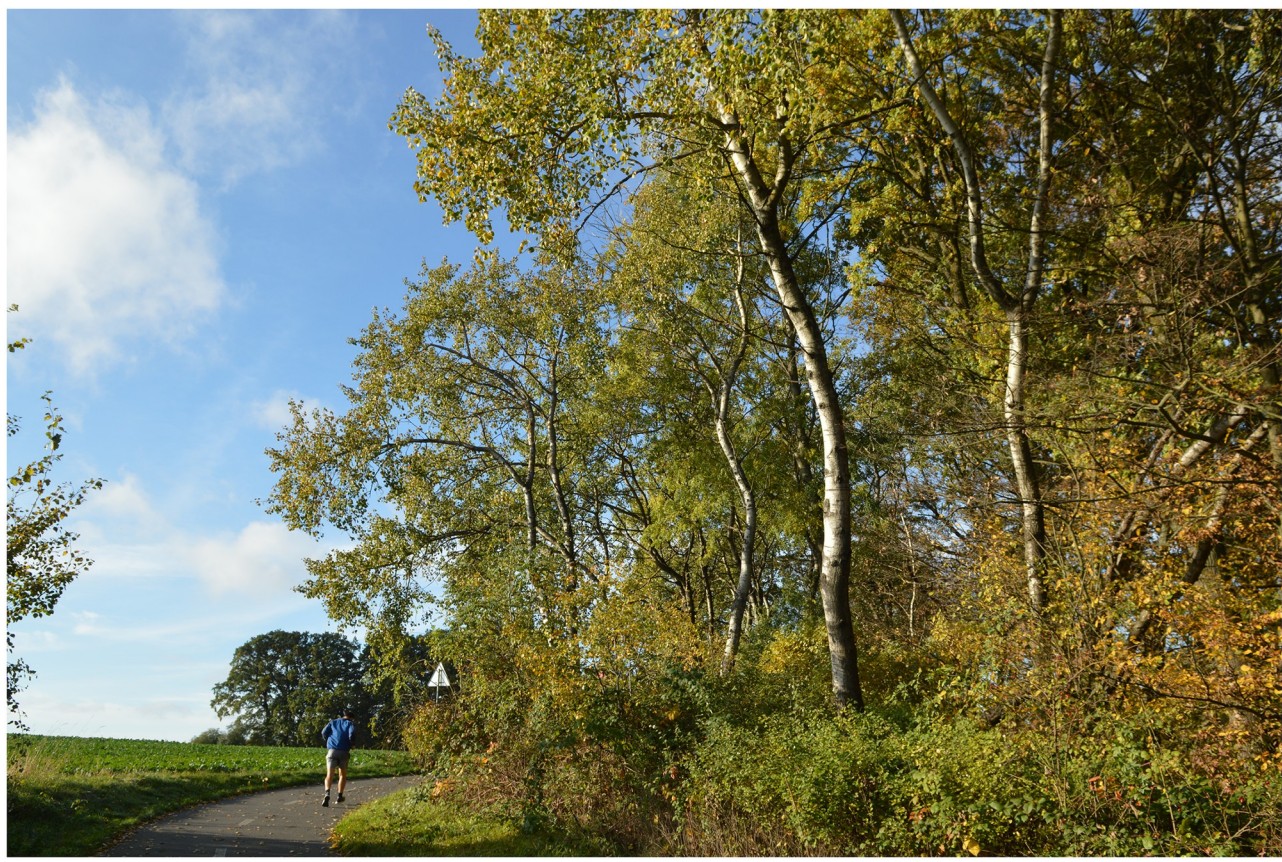

**Fig 1. Eurasian aspen (*Populus tremula* L.).** An often picture of low-quality aspen at the forest edge trying to expand to a meadow/field but prevented by every year activities such as haymaking or plowing, Beskydy Mnts., Czech Republic, Europe. Reprinted from the personal photo archive under a CC BY license, with permission from Antonín Kusbach, original copyright 2015.

mainland Europe, central Asia, and Siberia. In the South, aspen reaches Spain, Italy, Algeria, Greece, and Turkey [6, 12, 13]. A huge geographical span also occurs in North American aspen (*Populus tremuloides* Michx.) [5, 8] (Fig 2). In Europe, aspen naturally coexists with many broadleaved species. Contrary to North American aspen, its overstorey dominance is scarce with exception of early successional disturbed areas such as former quarries, excavations, clearings, brownfields or forest edges where it creates nearly uniform low timber quality groves (Fig 1). As an anemophilous tree, it produces billions of tiny seeds, but regenerates mostly by root suckers. Aspen populations form single living organisms called clones where "trees" are genetically identical and linked by expansive root systems [8, 14, 15]. In Europe, we can find dominant aspen in natural sub-boreal forests, where it colonizes vast disturbed areas [6, 12, 16]. While vegetative reproduction and r-strategy (a first tree colonist, prolific reproduction—seral species, e.g., [17]) is common for other tree species in central Europe (*Tilia*, *Betula* spp., *Corylus avellana* etc.), a rapid post-fire regeneration advantage by suckering is common in aspen. However, long-term single-species dominance by aspen is more common in North America [4, 18]; though it also appears to occur in eastern Europe [19]. In central Europe, aspen coverage is relatively sparse, despite its global distribution [20, 21].

Natural distribution of aspen in Europe is difficult to ascertain due to intense human impacts and active denudation of aspen communities over several centuries [5, 17, 20, 22]. For example, aspen and common birch were disseminated as amelioration species (i.e., able to

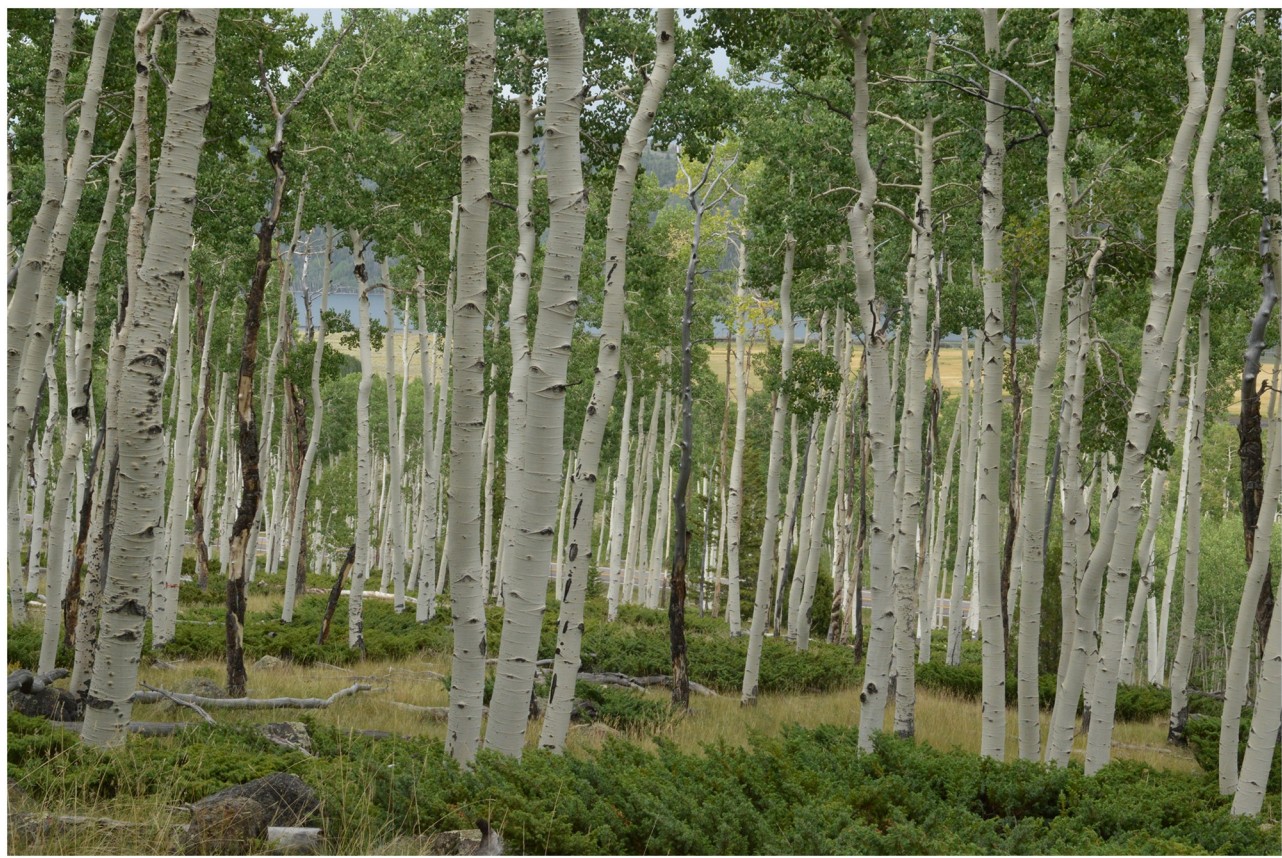

**Fig 2. North American aspen (*Populus tremuloides* Michx.).** A stable aspen clone, Pando, Fish Lake, Utah, USA. The aspen presence has been confirmed for 9 000 years at the place [14]. Reprinted from the personal photo archive under a CC BY license, with permission from Antonín Kusbach, original copyright 2015.

improve an extremely acidic soil environment after the massive decline of Norway spruce monocultures due to the acid rain in CZ [23]). Production and export of match sticks relied on establishment of aspen plantations during the 1970s in the former Czechoslovakia. Since this period, the interest in aspen timber and related research terminated. Some of these plantations have gone unmanaged, now becoming "naturalized" forests. Such surviving aspen forests opportunistically may spawn healthy communities with prospects of greater diversity, climate change resilience, carbon sequestration, enhanced litter chemistry, and other ecological services [8].

Noncommercial services provided by aspen in forest ecosystems exceed use of aspen wood and timber. Aspen quickly stabilizes denuded sites (shallow rocky soils, steep slopes), controls soil erosion [5, 6, 12], and appears to sequester a greater soil carbon amount than conifer forests [24]. Aspen also provides domestic and wild ungulates with high nutrition as forage [4]. A great number of organisms depends on it as a keystone species [5, 7, 8, 17, 25, 26]. Moreover, aspen forests are known to resist air-polluted and enhance microclimate [12, 27]. Finally, aspen is considered an "amenity species" for the added aesthetic dimensions its' golden leaves lend to forest vistas in autumn [4, 7].

Previous work has shown that Eurasian aspen can tolerate extremely diverse habitats [17] and depict aspen's coarse- (macroclimate, geography) and fine- scale (geology, topography and soils) ecological plasticity (20, 13). On the other hand, a historically low interest in cultivation disqualified aspen from regular forest management, resulting in coverage declines in the species [9]. To date, we know of no data driven study supporting a geographical extent, ecological niches, or ecological functions of *P. tremula* forest communities [*sensu* 28].

A focused understanding of European aspen's optimum settings, as well as sound stewardship of contemporary aspen, will improve regional biodiversity at-large due to the keystone role this species in known to play [8, 17]. Specific research goals include: (i) to assess important ecological characteristics associated with the aspen's broad ecological amplitude, (ii) to delineate the ecological and growth optimum (niche) of aspen in central Europe, and (iii) to examine and summarize reasons for the aspen's current distribution and potential paths for improving the aspen habitat regionally.

## Methods

### Data sources

The full territory of CZ (ca 78,000 km$^2$) was the study area. We used the forest management plan (FMP) and guideline (FMG) database for the CZ forest land (over 26,000 km$^2$); a source of specific data within the study area valid by the end of 2019. This database encompassed 4,656,130 stand groups (SG) described by the field mensuration. The character of the data is determined by established legislative protocols (FMP on a forest property > 50 ha, FMG on a forest property < 50 ha; Regulation No. 84/1996 Sb.) and originates from dendrometric measurements (e.g., diameter breast height, tree height, basal area, stand volume etc.) and formal management plans. We used data elements filtered from the entire database where aspen was present (cover > 1%) for each SG area. Additionally, we reduced this dataset by dropping aspen plantations (i.e., artificial stands established in 1970s for the match industry). The final raw aspen dataset encompassed 91,637 SG/rows.

### Aspen's ecological characteristics and niche

For each SG, we calculated (i) general climatic characteristics—seasonal air temperatures (mean, max, min) and precipitation using the procedure available at: http://worldclim.com, and (ii) geomorphometric indices for topographic variables based on a 30 m pixel digital

terrain model [29]. These indices reflected topography and soil moisture affected by local climate conditions such as a potential thermal inversion, and solar radiation exposure (slope, aspect) [30–32] (S1 File). Next, we used the GIS LES_OPRL layer [33]. This layer demarcated forest from non-forest and allowed calculation of aspen vs. other SG disturbance-management relations. The QGIS software and System for Automated Geoscientific Analyses (SAGA, v. 6.2.0) was used in the calculation of the indices. Additional categorical factors with formal information for SGs with aspen present were the forest ownership, forest extent, and forest category defined by the forestry legislation (Act on Forests No. 289/1995 Sb.; Regulation No. 298/2018 Sb.; Regulation No. 84/1996 Sb.). These factors described anthropogenic disturbances (e.g., active/passive, intensity, commercial, conservative etc.; S1 Table). The climatic, geomorphometric, and categorical characteristics were *explanatory factors* in subsequent analyses.

To introduce ecological characteristics, presence, and distribution, we displayed aspen's growth (i.e., productivity, represented by a site index [mean tree height]), and the area of an SGs' cover in an ecological/edatopic grid [34]. The aspen site index and SG aspen area were considered *response variables* for this work. The Czech Forest Ecosystem Classification (CFEC) [35, 36] is formally expressed by the ecological grid. This system orders forest site conditions along major environmental gradients; climatic and moisture-nutrient. For this study, we express the climatic gradient (altitudinal climate) by potential natural vegetation communities (*vegtypes*) to get a floristically homogeneous climatic framework. These vegtypes were asserted by climatic climax species of *Quercus*, *Fagus*, *Abies-Fagus*, and *Picea* [35, 37]. The moisture-nutrient gradient was represented by topo-edaphic units (*ecoseries*). These were xeric/extreme (X), poor (P), rich (R), humus (H), influenced by the fluctuating ground water table, rich (WR) and influenced by the fluctuating ground water table, poor (WP). Both vegtypes and ecoseries were used in the follow-up ecological grid and analytical classes.

The aspen productivity and cover were displayed as dissemination heatmaps with average productivity (site index) and area values of the classes in each cell of the ecological grid using the aspen presence data (91,637 SGs with the aspen proportion > 1%). We assumed that the productivity-based heatmap reflected a greater environmental (climatic, site-specific) signal compared to management/disturbance responses. The area based heatmap should address the opposite; intense management changed the area of the aspen presence. To compare aspen characteristics with other forest stands we constructed the same global heatmaps for all CZ forests. The classes were visualized and ordered to highlight a pattern of dissimilarity using a dendrogram scaling function [38]. This procedure computed the Euclidean distance between both rows and columns, the pattern represented the environmental dissimilarity of the classes. For heatmap construction, we used the "heatmaply" package [39].

## Aspen's environmental and spatial analysis

To identify ecological gradients associated with important environmental characteristics of aspen, we used principal component analysis (PCA). We further reduced the aspen dataset by dropping SGs with the aspen proportion < 50% of each SG area for improving accuracy. First, we used 11,366 aspen-dominant SGs and 54 factors. This set was further reduced to 6,157 SGs and 31 factors to reduce data (software technical limitations) and statistical noise caused by unrelated factors. Factors accompanying randomly chosen SGs were presented in Table 1. We transformed factors with |skewness| > 1 and checked the dataset for outliers [40]. Data were normalized based on standard deviations. After an orthogonal rotation optimization, we get independent, mutually uncorrelated principal components (PC). Significance of the PC were

**Table 1. Factors used in the analysis.**

| Ecological units | Abbreviation | Units/Values |
|---|---|---|
| Vegetation types | vegtypes | categorical/1–4 |
| Ecological series | ecoseries | categorical/1–6 |
| **Ownership, forest property size, categorization***  | | |
| Forest ownership | owner | categorical/1–5 |
| Size of the forest property (FMG < 50 ha, FMP > 50 ha) | size | categorical/1–2 |
| Forest category* | categ | categorical/1–6 |
| **Climatic factors** | | |
| Spring Mean Precipitation | prec_spr | mm/115–218 |
| Summer Mean Precipitation | prec_sum | mm/200–395 |
| Fall Mean Precipitation | prec_fall | mm/98–299 |
| Winter Mean Precipitation | prec_win | mm/64–287 |
| Spring Mean Temperature | t_spring | ˚ C/2.7–10.4 |
| Summer Mean Temperature | t_summer | ˚ C/11.7–19.5 |
| Fall Mean Temperature | t_fall | ˚ C/4.0–9.9 |
| Winter Mean Temperature | t_winter | ˚ C/-4.6–1.0 |
| Spring Mean Max Temperature | tmax_spr | ˚ C/6.3–15.6 |
| Summer Mean Max Temperature | tmax_sum | ˚ C/15.8–25.3 |
| Fall Mean Max Temperature | tmax_fall | ˚ C/6.9–14.1 |
| Winter Mean Max Temperature | tmax_win | ˚ C/-2.3–3.7 |
| Spring Mean Min Temperature | tmin_spr | ˚ C/-0.8–5.6 |
| Summer Mean Min Temperature | tmin_sum | ˚ C/7.5–14.2 |
| Fall Mean Min Temperature | tmin_fall | ˚ C/1.2–6.2 |
| Winter Mean Min Temperature | tmin_win | ˚ C/-6.9–-1.6 |
| **Topographic/geomorphic factors** | | |
| Altitude | alt | meters/141–1154 |
| Aspect value | av | 0–1 |
| Slope | slope | degrees/0-44 |
| Terrain shape | t.shape | categorical/1-3 |
| Landform topography | landf TP | categorical/0-9 |
| Direct insulation | Direct I | values 3.7–7.3 |
| Diurnal anisotropic heating | Diurnal | values -0.58–0.54 |
| Multiresolution Index of Valley Bottom Flatness | MRVBF | values 0–7.97 |
| Negative Openness | negative | values 1.14–1.59 |
| Protection | Protecti | values 0–0.43 |
| Texture | Texture | values/0–100 |
| Topography position index | TPI | values/-24.28–20.22 |
| Topography wetness index | TWI | values/4.16–13.12 |
| **Mensuration (response) factors** | | |
| Site index | si | meters |
| Area of a stand group (SG) | area | ha |

* Act on Forests No. 289/1995 Sb.; Regulation No. 298/2018 Sb.; Regulation No. 84/1996 Sb. Categorization is aggregated for the purpose of the study

tested using a Monte Carlo randomization test with 1000 runs. We calculated the linear Pearson's *r* and rank Kendall's *tau* correlation coefficients as the relationship between the ordination axes and factors, using a threshold of the coefficients > 0.4. We displayed the PCA analytical classes into the ordination environmental space using the PC-ORD 6 software [40].

We tested the analytical classes with the Random Forests supervised discrimination (RF) [41] to: (i) discriminate among the classes, and ii) identify factors that were significantly associated with the PC. These factors were ranked in the RF variable importance function according to Mean Decrease Accuracy (MDA) [42]. The best RF solution was revealed by the lowest "out-of-bag" estimate of the error rate as a measure of a general RF misclassification [43]. For the relevant classes, we calculated significant limits using the "prototype" function. We used R —version 4.0.3 [44] for the RF analysis.

Next, we modeled spatial relations of the aspen SGs with global forest SGs using the GIS LES_OPRL layer as a calculation of a distance between a SG centroid (for the aspen presence dataset > 1%, dominant aspen dataset > 50%, and a random selection of a global forest dataset) and a closest, geographically defined point of the border between a forest and non-forest using the QGIS software. We verified the calculated distance using the F-test and graphed aspen SGs and global forest SG differences.

Finally, we modeled the aspen proportion and stand area, environmental (most significant site-specific), and management (forest ownership, forest property size and categorization, explaining a type and intensity of management) factors for the aspen-dominant data (11,366 SGs with the aspen proportion > 50%). To test the effects of those complex interlinkages on aspen performance, we employed the Generalized Linear Models (GLM) with the Gaussian error distribution [45]. All explanatory factors were standardized to a zero mean and SG variance. For each GLM model, we calculated $R^2$ using the "MuMIn" package version 1.42.1. All the analyses were carried out in R version 4.0.3 [44].

## Results

### Describing aspen's broad ecological amplitude with key characteristics

Based on the joint FMP and FMG database, the aspen proportion was 0.28% as a simple ratio of the total forest area/total aspen area in CZ. The mean SG area for all FMP forests was 1.49 ha, while 0.47 ha for FMG forests. The mean SG area for all forests was 1.02 ha, while for admixture aspen SGs (> 1%) it was 1.04 ha and aspen dominated SGs (> 50%) 0.20 ha.

Aspen covered a broad range of ecological conditions represented by the mean productivity and mean SG area at the heatmaps (Figs 3–6). The only gap within this ecological grid appeared at the *Picea*/Humus and *Picea*/Rich units. There were too little or no data (*Picea*/Humus sites do not exist) for these units. The aspen heatmap (Fig 3) showed the greatest productivity in the *Quercus*, *Fagus* and *Abies-Fagus* communities/vegtypes, and on mesic sites. The productivity of aspen on the vegtype/macroclimatic gradient was relatively flat showing minimal productivity in the *Picea* vegtype. The ecoseries productivity development was better pronounced, revealing a clear difference between the water affected sites and the rest of the heatmap. The lowest aspen productivity applied to the xeric sites (Fig 3). The global forest heatmap displayed the best overall productivity in the *Fagus* and *Abies-Fagus* vegtypes with a visual signal in ecoserial moisture-trophic development (Fig 4). The aspen area heatmap showed the greatest mean SG area in the *Picea* communities and on the xeric sites, which was different from the rest of the heatmap area (Fig 5). The global forest heatmap displayed the same pattern across all vegtypes (Fig 6).

### Environmental factors define optimum habitat niche

The PCA ordination of the aspen dataset revealed four significant PC (p = 0.001). PC1 through PC4 were explaining 79% (46, 15, 10, 8 respectively) of the data variance. We interpreted PC1 as a temperature gradient being strongly associated with seasonal temperatures (MDA t_spring: r = -0.98, tau = 0.87; t_summer: r = -0.96, tau = -0.83) and altitude as a general

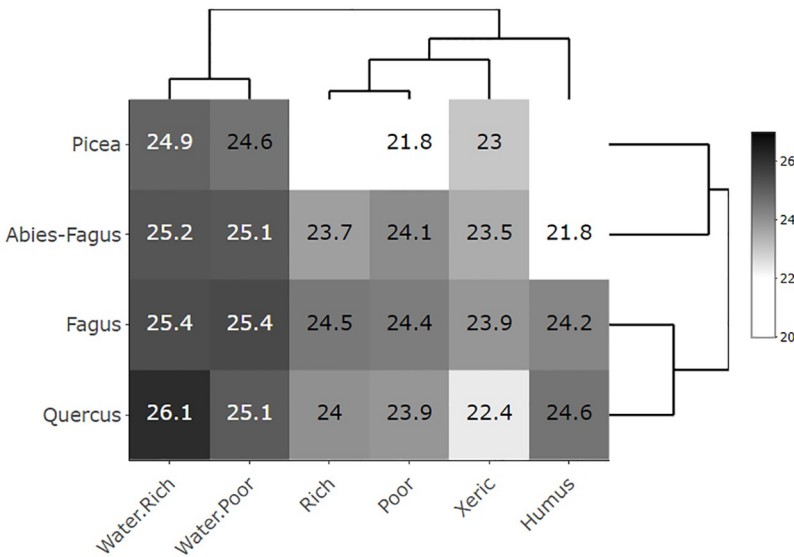

**Fig 3. Eurasian aspen (*Populus tremula* L.) productivity heatmap.** The heatmap's values demonstrate mean tree heights/site indices in a scale of 20–27 meters representing productivity of aspen (N = 91,637) in the Czech Republic depending on the elevational/macroclimatic and moisture-fertility gradient. The vegtypes in rows represent the elevational gradient and ecoseries in columns represent the moisture-fertility gradient. Both gradients forming an ecological grid [34] were visualized and ordered by dendrogram scaling.

climatic proxy (alt: r = 0.87, tau = 0.66). PC2 was interpreted as a topographically driven moisture gradient being strongly associated with topography (MDA slope: r = 0.84, tau = 0.66; Negative: r = -0.76, tau = -0.58; MRVBF r = -0.72, tau = -0.57). PC3 was suggested a precipitation gradient driven by seasonal rain and snow (S2 Table). Summarized, we found the climatic

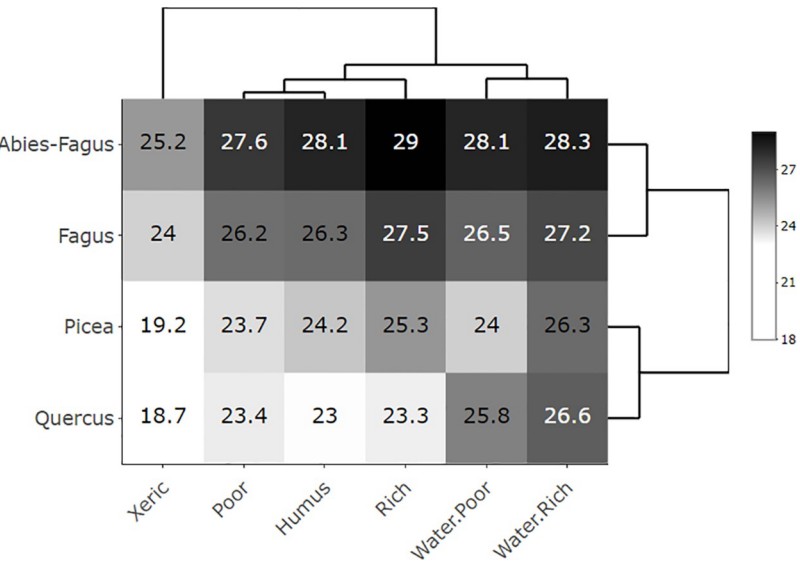

**Fig 4. Global forest productivity heatmap.** The heatmap's values demonstrate mean tree heights/site indices in a scale of 18–28 meters representing productivity of no-aspen forests (N = 2,523,687) in the Czech Republic depending on the elevational/macroclimatic and moisture-fertility gradient. The vegtypes in rows represent the elevational gradient and ecoseries in columns represent the moisture-fertility gradient. Both gradients forming an ecological grid [34] were visualized and ordered by dendrogram scaling.

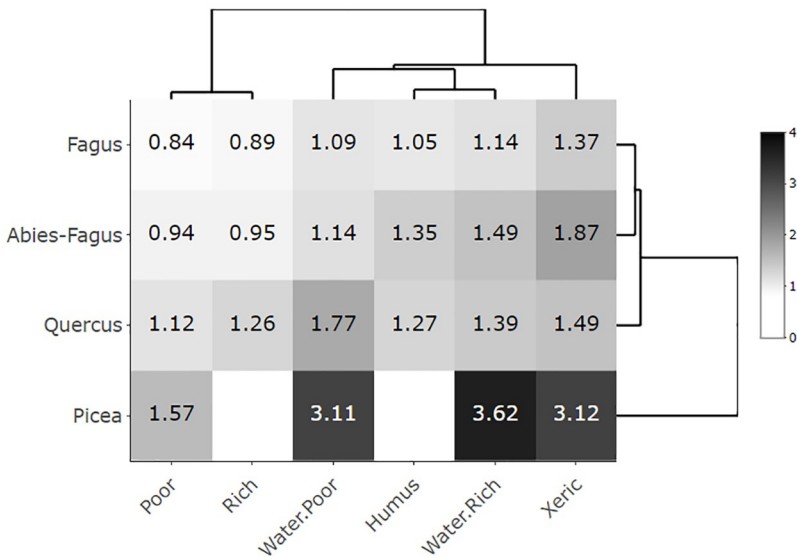

**Fig 5. Eurasian aspen (*Populus tremula* L.) area heatmap.** The heatmap's values demonstrate mean stand group areas in hectares of aspen (N = 91,637) in the Czech Republic depending on the elevational/macroclimatic and moisture-fertility gradient. The vegtypes in rows represent the elevational gradient and ecoseries in columns represent the moisture-fertility gradient. Both gradients forming an ecological grid [34] were visualized and ordered by dendrogram scaling.

gradient and terrain topography to be the main environmental drivers of the aspen presence. The response factors appeared to be insignificant in the PCA ordination (site index: r = 0.09, tau = 0.04; area of a stand group: r = -0.04, tau = -0.02). While the visualization of the vegtypes into the PC ordination space described potential communities well by showing climatic/temperature gradient development, the ecoseries display was less conclusive showing obscure climatic, moisture, or even fertility gradient development (Fig 7, S1 Fig).

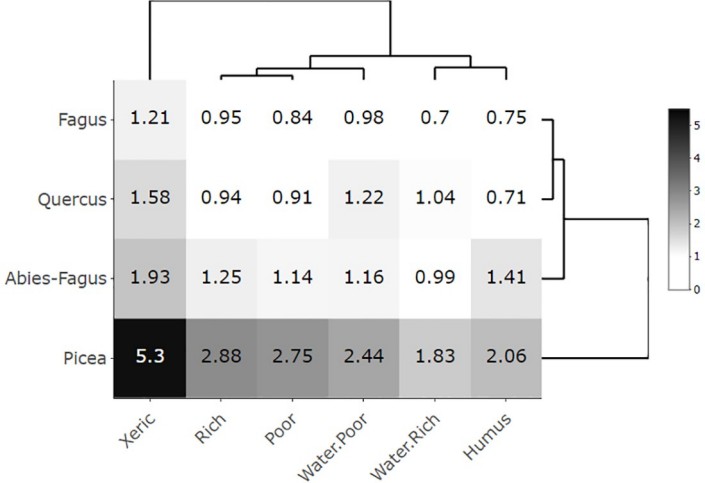

**Fig 6. Global forest area heatmap.** The heatmap's values demonstrate mean stand group areas in hectares of no-aspen forests (N = 2,523,687) in the Czech Republic depending on the elevational/macroclimatic and moisture-fertility gradient. The vegtypes in rows represent the elevational gradient and ecoseries in columns represent the moisture-fertility gradient. Both gradients forming an ecological grid [34] were visualized and ordered by dendrogram scaling.

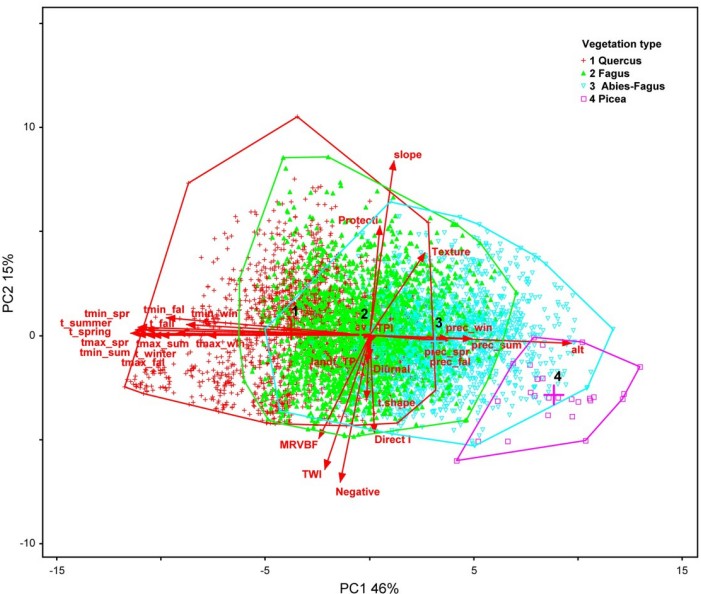

**Fig 7. The principal component analysis ordination of the aspen data set.** An ordination biplot of aspen stand groups (N = 6,157) presents the most influential gradients PC1 and 2 on the axes, the most influential factors as the red vectors and the vegetation type envelopes. For the vector labels, see the Table 1.

The RF classification analysis revealed a 15% misclassification rate for the discrimination of the vegtypes. Important site-specific environmental factors influencing the vegtypes discrimination were identified by order of importance in MDA: the climatic factors–altitude (99.1), prec_-spring (83.0), the calculated geomorphometric indices–MRVBF (58.5), TPI (51.4), TWI (43.2), and two directly easy-measurable terrain characteristics–slope (33.9) and landform topography (19.9). For the vegtypes, we calculated significant environmental characteristics (Table 2).

The RF results were consistent with the results of PCA. A dominant role of climate suggested by RF corresponded with the climatic gradient of PC1. While the moisture gradient (PC2) was significant in PCA, RF did not prove the environmental factors to be effective in ecoseries (topo-edaphic) discrimination. The important environmental (site-specific) factors revealed in the RF analysis along with forest management/anthropogenic disturbance factors were used in further modeling.

## Aspen occurrence today: Spatial, proportional, and functional relations

The spatial analysis showed a significant difference among distances of SGs from the forest–nonforest boundary. The distance median/mean was 18/34, 32/70 and 42/93 m for the aspen

**Table 2. Environmental characteristics of the vegtypes.**

| Vegtype | Altitude | Slope | Landform | MRVBF | TPI | TWI | Prec_spring | Tmin_winter |
|---|---|---|---|---|---|---|---|---|
| | m a.s.l. | deg | | | | | mm | ° C |
| *Quercus* | 187 | 0 | 6 | 58.6 | 0.0004 | 10.1 | 118 | -2.8 |
| *Fagus* | 421 | 5 | 3 | 1.26 | -2.4 | 8.01 | 131 | -3.6 |
| *Abies-Fagus* | 633 | 8 | 5 | 0.3 | -0.95 | 6.83 | 145 | -5.2 |

Note: for the factor labels, see the Table 1.

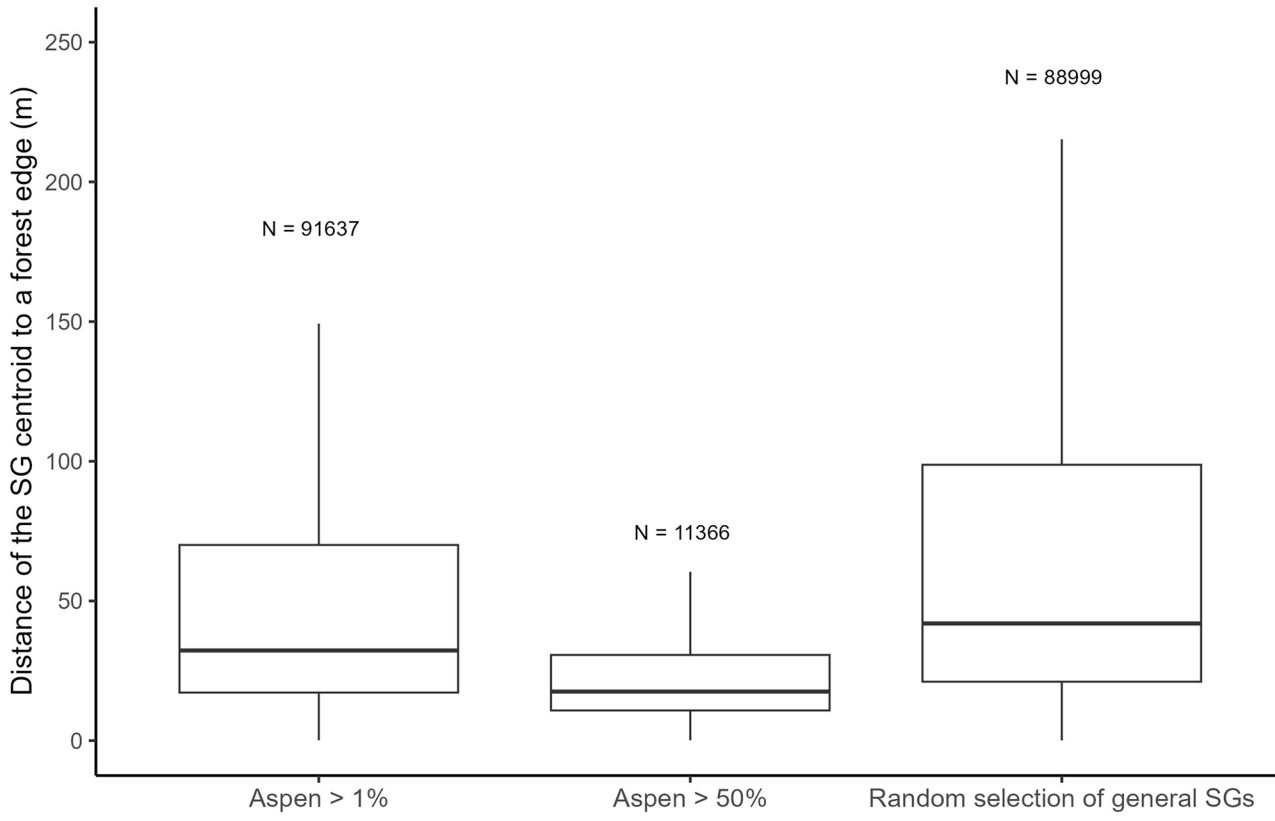

**Fig 8. Distance of a stand group (SG) to a forest–nonforest boundary.** The spatial analysis of the aspen and global/general forest SGs showed a significant difference between their distances from the forest–nonforest boundary. F-test between aspen datasets: F = 2.899, p-value < 2.2e-16. F-test between the aspen > 1% and global forest: F = 0.623, p-value < 2.2e-16. F-test between the aspen > 50% and global forest: F = 0.215, p-value < 2.2e-16.

dominant, aspen presence, and global forest SGs, respectively (Fig 8). After the 11-step-by-step reductions in the number of analytical factors, the GLM models revealed that the response of the aspen productivity for the most influential site-specific factors was weak ($R^2_{adj}$ = 0.069). However, we found significant association between the proportion of aspen and the size of forest property expressed by FMG, FMP, and the forest category (Fig 9).

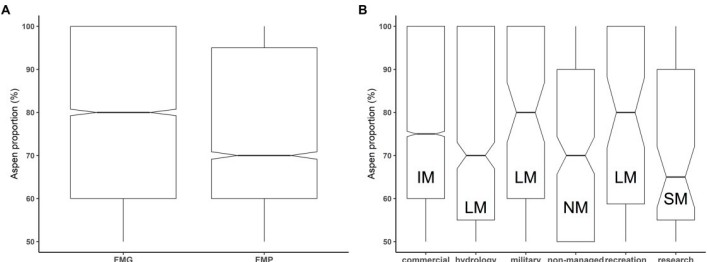

**Fig 9. Proportion of aspen in aspen-dominant stand groups.** Association of the proportion of aspen with (A) the size of a property expressed by Forest Management Guidelines (FMG) for properties < 50 ha, and Forest Management Plans (FMP) for properties > 50 ha, (B) the forest category, IM = intensive, SM = specific, LM = low, NM = no management (S1 Table).

## Discussion

### Aspen ecological potential in central Europe

Both Eurasian and North American aspen are known as species with the enormous ecological amplitude [8, 17, 46]. Wide ecological variance and infrequent appearance of aspen have been recognized for a long time [9, 47–49]. In our study, aspen presence was clear via a simple statistic; the aspen proportion of 0.28% on the total forest area in CZ was negligible. Based on the Landscape Inventory CzechTerra, the aspen proportion across all CZ forests was 0.7% (www. czechterra.cz/#2015). Both surveys describe the species as rare. For that reason, aspen has not been inventoried independently in the National Forest Inventory, but consistently grouped with other softwoods such as lime tree, willows, and other poplars for 4.6% [50]. Stands with dominant aspen (> 50%) are sporadic and mostly very small (mean = 0.2 ha). The total area of these stands is considerably lesser than stated by Worrell [5]. The wider coverage of this species was obvious from a simple display of aspen stands in the territory of CZ where it was found everywhere from the lowest to high mountain elevations (Fig 10), exhibiting clear environmental adaptability.

The broad ecological potential of aspen was clear from the heatmaps (Figs 3 and 5). Aspen was present on a wide range of sites, from very dry (sand dunes or screes) to quite wet (waterlogged, peats) and among both poor and rich soils [17]. Our findings, proved on site-extensive data, were consistent with the empirical knowledge of Vincent [51], Chmelař [52], and Úradníček [53]. Besides the clear environmental potential for the widespread aspen habitat, other relevant signals included aspen growth and distribution. We also observed a pattern of consistent aspen performance in (i) xeric sites and (ii) *Picea* communities. This display of the aspen-conducive attributes might indicate additional controls on aspen growth on both uncommon and common sites (i.e., the rest of the heatmap where aspen performance is relatively even; Fig 5).

The PCA ordination and RF classification of an array of environmental and mensuration data on aspen dominant sites confirmed the broad ecological amplitude and plasticity suggested by the heatmaps. While aspen is traditionally reported up to 800 m elevation and higher [54], in central Europe, we found it can grow on sites from ca 100 m up to 1200 m a.s.l. (Fig 11, authors' observations). Aspen distribution represents an enormous gradient of 9.6, 7.8, 5.5˚ C in annual mean, maximal, and minimal temperatures, respectively, between the warmest and coldest SGs, and almost 1000 mm in precipitation differences between the most and least rainy SGs. The climatic/temperature gradient was the most significant in all analyses and represented by the climatic proxy–altitude (Fig 11). The vegtypes and associated significant climatic factors (Fig 7; S2 Table) represented potential natural vegetation (PNV) *sensu* Tüxen [55] or zonal/climatic climax *sensu* [37, 56] despite the fact they were analyzed on aspen dominant stands. This evidence excluded aspen from the PNV and the zonal/climatic climax concept [57] confirming aspen as a generalist and seral species in central Europe (Figs 10 and 11).

### An expansive ecological niche suggests neglect and undervaluing of a keystone system

Ecological and growth optima, using a climate proxy represented by elevational gradients, are known for Norway spruce [58, 59] and European beech [60] in central Europe. In the case of spruce, these two optima vary because the species was introduced outside its natural range [1] and it is more productive in lower novel locales. Aspen, as a generalist, can grow almost everywhere; nevertheless, it performs differently in varied ecological conditions (Figs 3, 5 and 7). Therefore, it is advantageous to delineate the preferred niche of aspen in the ecological grid

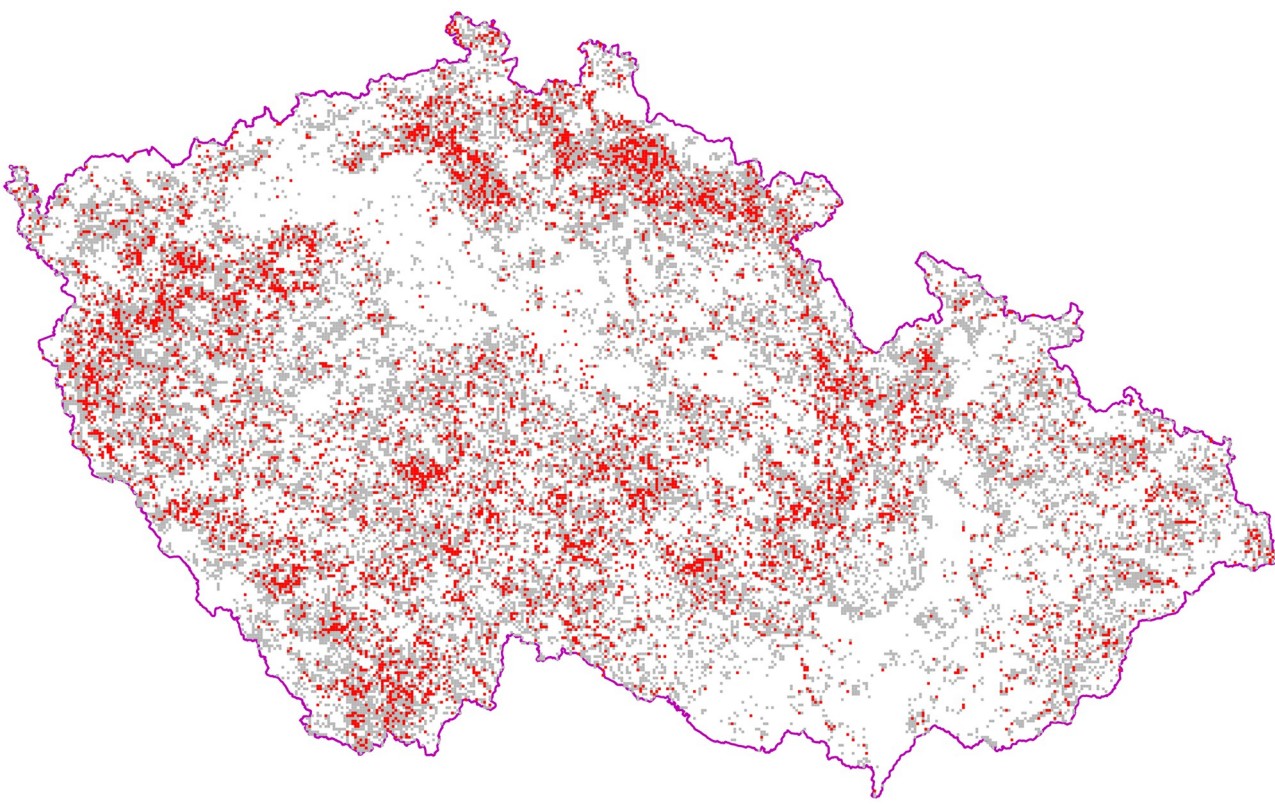

**Fig 10. Eurasian aspen (*Populus tremula* L.) distribution in the Czech Republic.** Squares 1 × 1 km represented aspen presence and stand groups with the aspen cover > 1% (gray squares) and > 50% (red squares). N = 91,637. Reprinted from the GIS analysis of the data under a CC BY license, with permission from Robert Hruban, original copyright 2021.

using a climatic optimum (elevation climate indirectly revealed by the vegetation communities–vegtypes) combined with site-specific characteristics (represented by the ecoseries). The aspen productivity heatmap showed the greatest affinity at the intersection of the *Quercus* and *Fagus* communities and ground water affected sites (Fig 12). Thus, based on our data, a

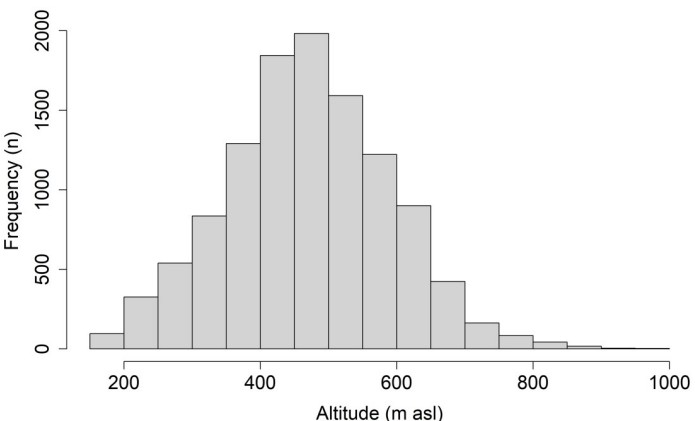

**Fig 11. Aspen altitudinal distribution in the Czech Republic.** A number of aspen dominant stand groups (n) was dependent on the Czech altitudinal gradient. N = 11, 366.

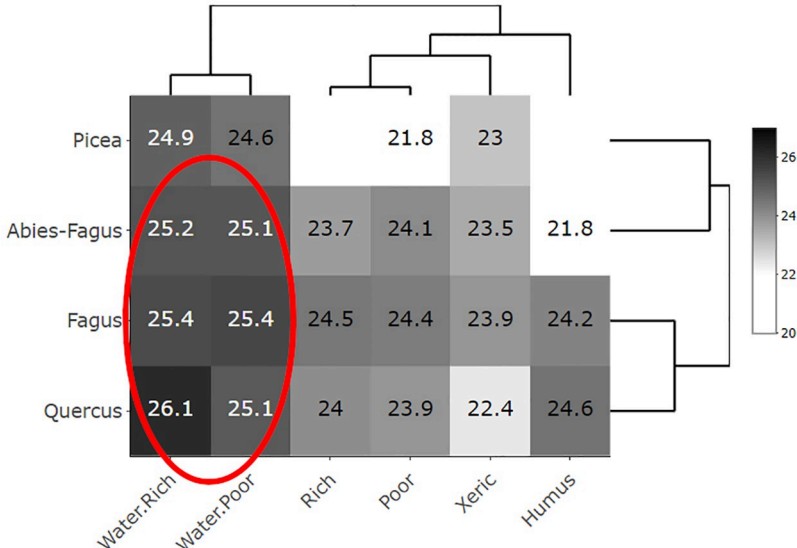

**Fig 12. Productivity heatmap with delineation of the Eurasian aspen (*Populus tremula* L.) ecological and growth optimum.** Aspen productivity is expressed by a site index, a mean height of an aspen tree in meters, in a scale of 20–27 meters. The vegtypes in rows represent the elevational gradient and ecoseries in columns represent the moisture-fertility gradient. Both gradients forming an ecological grid [34] were visualized and ordered by dendrogram scaling.

combination of the climate proxy and soil moisture provides the strongest growth driver of aspen (Figs 3, 5 and 7, S2 Table and S1 Fig). Looking at significant environmental characteristics, an aspen ecological niche (a.k.a., "realized niche" *sensu* [28]) can be climatically defined by altitude of 187–633 m asl (Fig 11), which corresponds with a spring precipitation of 118–145 mm and winter temperature minimum of -2.8 –-5.2˚ C. Favorable soil moisture is significantly associated with terrain; relatively broad, flat, and open areas (slope 0–8 degrees, landform 5, 6) and concave topography (landform 3, 4, MRVBF 0.3–58.6, TPI -2.4–0.0004, TWI 6.83–10.1) (Table 2). Still, species optima are spatially dependent, meaning their delineation may be different across the geographical spectrum. Pan-regional ecological structuring e.g., [38, 61] needs to be checked before optima are established.

The heatmaps, PCA and RF analyses combined the significant influence of environmental factors (altitudinal climate, soil moisture) on the aspen productivity and distribution. However, single response variables (site index and SG area) appeared mostly as statistical noise showing marginal association with the other environmental factors in our analyses. Moreover, a comparison of the ecological optimum of aspen (Fig 12) with the actual distribution of the dominant aspen SGs (Fig 5) and the patterns discussed above, irrespective of demonstrated environmental significance, suggested that aspen growth and distribution have been controlled not only by the environment, but also by human manipulations and other natural disturbances.

## Aspen past and present: Pathways for improved habitat

In central Europe since early 1800's, there has been little interest in aspen and other broad-leaved softwood species due to short-rotation profit-driven forestry [9] favoring fast-growing conifers (spruce, pine) often husbanded in monoculture settings [62, 63]. Based on the Saxon system of a "normal forest" of age classes (e.g., [64, 65]) aspen, birch, mountain ash, and

willows have been deliberately replaced from "cultural forests" as ubiquitous "weeds" or competitors inhibiting production forestry [1, 47, 48, 51, 62, 63].

Since the 1960s, aspen has been further curtailed from forests by intensive management (cleaning, thinning). In such "normal forests", surviving aspen SGs remained small and scarce (Fig 5). Our multi-functional models considering both environmental and management (anthropogenic disturbance) factors showed that both active (low-intensity, intensive) and passive (no action) management were of high importance in past aspen distributions. Traditionally limited, low-intensity, management on small forest properties (< 50 ha, controlled by FMG) has favored persistence of aspen (Fig 9A). Intensive exploitation of forests on large properties (vast commercial monoculture complexes controlled by FMP), using clear-fell practices significantly decreased aspen presence, stand size, and stand distance from forest–nonforest boundaries (Figs 5, 8 and 9A).

Forest category was also a statistically significant factor representing a type and intensity of management affecting presence and distribution of aspen. High elevation and xeric *Picea* forests (Figs 3–6) are an example of legally protective areas experiencing passive management for ca 50 years. These non-managed forests displayed a low proportion of dominant aspen (> 50%) (Fig 9B) although aspen presence (> 1%) was common in SGs (Fig 5). This is a result of combination of historic management of forests under passive management for ca 50 years but still carrying a legacy of 250 years of the Saxon-style management, eradicating pioneer species [64, 65] and restricting environmental conditions for aspen growth in high elevations and dry sites. Though aspen did not return to those protected sites because of (i) low presence and (ii) passive management, (i.e., low anthropogenic activity such as a ground scarification), and (iii) high numbers of browsing ungulates in Czech forests, which has limited aspen recolonization by both asexual and sexual reproductive modes [66]. A low proportion of dominant aspen SGs in research and educational forests such as the Training Forest Enterprise Masaryk Forest Křtiny close to Brno city (https://www.slpkrtiny.cz/) can be explained by the targeted experimental approach taken in these forests in the past. Low-intensity management in military, recreation and forests with the hydrological function was also favorable for aspen establishment (Fig 9B). A paradox could be seen in former commercial forests; these were intensively exploited for industrial purposes and now have a lasting aspen presence as thriving aspen forests under low-intensity management. There are a lot of small, often detached patches of dominant aspen SGs in abandoned fields/meadows, gravel excavations, quarries, and along old roads. Following spontaneous invasion of aspen, these stands were reassigned to a commercial designation. Such isolated aspen groves likely skewed the result of the general commercial category (Fig 9B).

Nowadays, the pioneer species are viewed in a new light as ecologically valuable component of forest ecosystems. Six species of aspen constitute a global network of keystone species creating huge diverse systems around the northern hemisphere [8]. These systems stabilize incredibly high landscape biodiversity [8, 15]. Aspen in central Europe portends a promising versatile species singly, as well as a refugium for many obligate species in a climatically unsure future [67]. In forest management, it is necessary to foster the overlooked concept of seral species facilitation so critical to obligate species and preservation of functional processes [3, 4, 17]. In practice, this means facilitating remnant aspen stands. It is not an easy task to change established forestry practices where we have been cultivating "nice and clean" conifer monocultures for centuries. Exploitation and balance of two-phased (at minimum) regeneration of stands; the first phase using seral/pioneer species and the second phase consisting of targeted late-succession species [68–70]. Finally, introduction of missing aspen may be accomplished via silvicultural practices and sowing techniques [51]. Where knowledge is lacking, it is useful to revisit forgotten practices of earlier foresters [23, 48], as well as closely monitored experimental

methods [4]. Climate change, alongside overexploited forests, has led to massive degradation of conifer plantations across central Europe. Practical knowledge of aspen ecology and growth characteristics—a species which has been "hiding in plain sight"—is of high importance for community ecology and noncommercial forest ecosystem services, as well as fostering resilient and pliable forests as we face changing climatic futures. Reestablishing aspen in central Europe provides a sound strategy for process-based forest restoration, conservation, and adaptive management.

## Conclusion

We used large-scale forest mensuration-based data to demonstrate the broad ecological amplitude in *P. tremula*. Our novel ecological niche approach employs numerous environmental variables to explain biogeographic optima in aspen forest communities of central Europe, which have otherwise been hidden (or ignored) due to their patchy existence and underappreciated value. Irrespective of local ecology (i.e., the realized aspen niche) this study confirmed past commercial expediency in forestry is responsible for broad-scale aspen suppression in central European forests. Aspen demonstrates a wide amplitude of habitat preferences, but curiously we found only small and isolated communities. Past management has clearly played a detrimental role for this keystone species; meaning that diverse plant and animal assemblages that thrive under aspen have likely followed a similar declining trajectory. Neither potential aspen habitat, nor its biodiversity value, are being taken full advantage of where conditions are evidently present for widespread proliferation, though they have been underutilized. Locations in the Czech Republic predominantly influenced by natural forces demonstrate aspen's persistence even as other more shade-tolerant species established and grew within aspen stands. The notion that aspen play *only* a seral or pioneer role must be questioned; a versatile species employs many strategies to thrive and expand.

Recent interest in sustainable forest management acknowledging the importance of seral species, including aspen, parallels progressive management which seeks to emulate natural process over engineered forests, at least in areas where resilient and semi-natural conditions are desired (e.g., designated forest reserves). Such process-based stewardship is favored in forest restoration, for instance, after clear cuts following recent broad spruce bark beetle dieback in central Europe. Additionally, widespread promotion of ecologically adaptive aspen communities is expected to support forest resilience broadly, as well as biodiversity conservation under anticipated warming climates accompanied by increased disturbance frequency and intensity. For the purposes of species preservation, an adaptive management approach holds promise in central Europe to realize the full ecological potential of this widely acknowledged keystone species. Our challenge is to have the foresight to envision forests of this region, under dynamic climate conditions, poised for future adaptation rather than rigidly clinging to the agricultural forest models that have led to ecological realignments, species losses, and occasional ecosystem failure.

## Supporting information

**S1 File. Geomorphometric indices for terrain topography and soil moisture related characteristics.**
(DOCX)

**S1 Table. Factors characterizing a type and intensity of forest management/anthropogenic disturbance.**
(XLSX)

**S2 Table. Principal component ordination: Pearson and Kendall correlations with ordination axes.**
(TXT)

**S1 Fig. The PCA ordination of the aspen data set.**
(TIF)

**S1 Data.**
(PDF)

**S2 Data.**
(PDF)

## Acknowledgments

This research was performed in cooperation with the program QK 1920328, "ZEMĚ"—Program of applied research of the Czech Ministry of Agriculture for 2017–2025: Complex solution of forest regeneration and silviculture in areas with fast and massive forest decline, MENDELU IGA-LDF-22-IP-020, Western Aspen Alliance and the USDI Bureau of Land Management (Grant # L21AC10369-00).

We would like to thank the Forest Management Institute, Czech Republic for providing and preparation of the data.

## Author Contributions

**Conceptualization:** Antonín Kusbach, Jan Šebesta.

**Data curation:** Robert Hruban, Pavel Peška.

**Formal analysis:** Antonín Kusbach, Jan Šebesta, Robert Hruban.

**Methodology:** Antonín Kusbach, Jan Šebesta.

**Software:** Antonín Kusbach, Jan Šebesta.

**Supervision:** Antonín Kusbach.

**Visualization:** Antonín Kusbach, Jan Šebesta.

**Writing – original draft:** Antonín Kusbach.

**Writing – review & editing:** Antonín Kusbach, Jan Šebesta, Paul C. Rogers.

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
