## [Decision Letter · Decision Letter 0]

29 Aug 2023

PONE-D-23-20092Eurasian aspen (Populus tremula L.): Central Europe’s keystone species ‘hiding in plain sight’PLOS ONE

Dear Dr. Kusbach,

Thank you for submitting your manuscript to PLOS ONE. After careful consideration, we feel that it has merit but does not fully meet PLOS ONE’s publication criteria as it currently stands. Therefore, we invite you to submit a revised version of the manuscript that addresses the points raised during the review process.

Two reviewers have commented on the text and found it almost ready for publication. 

Please make sure to follow their suggestions. 

In addition although I also find the text generally satisfactory, it  could need a careful edit to remove excess words and improve  language, and I suggest that a proper proof is performed before resubmission of the manuscript.

The legends are extremly short and more information about how to read graphs and tables is needed. Kindly go through each legend and add text so that each figure becomes self explanatory in the final paper. 

And on a minor note, I suggest you move the legends and place them together at  the manuscript; that will provide a  better overview of the manuscript text and its flow. 

I find that alll the figures have quite low resolution, and the font size appears far to small. This should be taken care of in the updated version of the manuscript.

We look forward to receiving your revised manuscript.

Kind regards,

Benedicte Riber Albrectsen

Academic Editor

PLOS ONE

3. We note that Figures 1A and 1B in your submission contain copyrighted images. All PLOS content is published under the Creative Commons Attribution License (CC BY 4.0), which means that the manuscript, images, and Supporting Information files will be freely available online, and any third party is permitted to access, download, copy, distribute, and use these materials in any way, even commercially, with proper attribution. For more information, see our copyright guidelines: http://journals.plos.org/plosone/s/licenses-and-copyright.

1. You may seek permission from the original copyright holder of Figure 1A and 1B to publish the content specifically under the CC BY 4.0 license.

4. We note that Figure 7 in your submission contain map images which may be copyrighted. All PLOS content is published under the Creative Commons Attribution License (CC BY 4.0), which means that the manuscript, images, and Supporting Information files will be freely available online, and any third party is permitted to access, download, copy, distribute, and use these materials in any way, even commercially, with proper attribution. For these reasons, we cannot publish previously copyrighted maps or satellite images created using proprietary data, such as Google software (Google Maps, Street View, and Earth). For more information, see our copyright guidelines: http://journals.plos.org/plosone/s/licenses-and-copyright.

1. You may seek permission from the original copyright holder of Figure 7 to publish the content specifically under the CC BY 4.0 license. 

Additional Editor Comments:

Two reviewers have commented on the text and found it almost ready for publication.

Please make sure to follow the suggestions by the reviewers.

In addition although I also find the text generally satisfactory, it could need a careful edit to remove excess words and improve language, and I suggest that a proper proof is performed.

The legends are extremly short and more information about how to read graphs and tables is needed. Kindly go through each legend and add text so that each figure becomes self explanatory in the final paper.

And on a minor note, I suggest you move tue legends to the end of the manuscript, so you peremit better overview of the manuscript text and its flow.

All figures have quite low resolution, and the font text is far to small. This should be taken care of in the updated version og the manuscript.

Reviewers' comments:

Reviewer's Responses to Questions

**Comments to the Author**

1. Is the manuscript technically sound, and do the data support the conclusions?

Reviewer #1: Yes

Reviewer #2: Yes

2. Has the statistical analysis been performed appropriately and rigorously? 

Reviewer #1: Yes

Reviewer #2: Yes

3. Have the authors made all data underlying the findings in their manuscript fully available?

Reviewer #1: Yes

Reviewer #2: Yes

4. Is the manuscript presented in an intelligible fashion and written in standard English?

Reviewer #1: Yes

Reviewer #2: Yes

5. Review Comments to the Author

Reviewer #1: Traditionally, aspen has received undeservedly little attention in both forestry and nature conservation. This paper fills this knowledge gap and demonstrates the importance of aspen for sustainable forest management and biodiversity conservation. The paper is based on accurate and extensive quantitative research. The authors used modern methods of data analysis appropriate to the task. The reliability of the conclusions is therefore beyond doubt. The paper makes a significant contribution to forest ecology and will be of interest to a wide range of readers with an interest in sustainable forest management and biodiversity conservation.

In my opinion, this paper is of a high scientific standard and is fully appropriate for PLOS ONE. I have no serious comments on the substance of the study, the presentation of the results and their visualisation.

Reviewer #2: Authors analysis the measured data and describe the ecological characteristics and geographical distribution of Eurasian aspen in the Czech Republic, as well as reveals the reasons for its low prevalence in Central Europe and elucidates the importance of the species in forest restoration. The manuscript has fluent language, reasonable structure, and a solid data foundation, which is a very interesting paper. I have just some minor suggestion:

(1) Many paragraphs in the results section. It is recommended to classifiy them and write under subtitles.

(2) What does the value in Figure 2 and Figure 3 mean? please explain them.

(3) It is recommended to retain 2-3 decimal places for the F and P values in the title of Figure 5.

6. PLOS authors have the option to publish the peer review history of their article (what does this mean?). If published, this will include your full peer review and any attached files.

Reviewer #1: **Yes: **Natalya Ivanova

Reviewer #2: No

---

## [Author Response · Author response to Decision Letter 0]

30 Oct 2023

We followed all the suggestions/instruction of the Editor.

• We edited the text in terms of excess words and the language. Please see the file ‘Revised Manuscript with Track Changes’.

• We extended the legends properly. 

• The legends were moved to the end of the manuscript however, it is recommended by the Plos One journal to place them right after they appear for the first time within the manuscript text.

• We remade the figures for the higher resolution, 300 dpi minimum and the font for a larger size. 

The Reviewers’ comments.

There have been ‘no serious comments on the substance of the study, the presentation of the results and their visualisation’ by the Reviewer 1.

We followed all the comments provided by the Reviewer 2.

(1) Many paragraphs in the results section. It is recommended to classify them and write under subtitles. DONE.

(2) What does the value in Figure 2 and Figure 3 mean? please explain them. DONE.

(3) It is recommended to retain 2-3 decimal places for the F and P values in the title of Figure 5. DONE.

---

## [Decision Letter · Decision Letter 1]

6 Feb 2024

PONE-D-23-20092R1Eurasian aspen (Populus tremula L.): Central Europe’s keystone species ‘hiding in plain sight’PLOS ONE

Dear Dr. Kusbach,

Thank you for submitting your manuscript to PLOS ONE. After careful consideration, we feel that it has merit but does not fully meet PLOS ONE’s publication criteria as it currently stands. Therefore, we invite you to submit a revised version of the manuscript that addresses the points raised during the review process.

We look forward to receiving your revised manuscript.

Kind regards,

Carlos Rouco, PhD

Academic Editor

PLOS ONE

Journal Requirements:

Additional Editor Comments:

Dear authors,

It seems that the manuscript has already undergone a second revision with very positive comments from both reviewers. In fact, one of them recommends its acceptance. Therefore, I recommend that the article be accepted after the authors carry out the minor revisions suggested by one of the reviewers.

Reviewers' comments:

Reviewer's Responses to Questions

**Comments to the Author**

1. If the authors have adequately addressed your comments raised in a previous round of review and you feel that this manuscript is now acceptable for publication, you may indicate that here to bypass the “Comments to the Author” section, enter your conflict of interest statement in the “Confidential to Editor” section, and submit your "Accept" recommendation.

Reviewer #1: All comments have been addressed

Reviewer #2: (No Response)

2. Is the manuscript technically sound, and do the data support the conclusions?

Reviewer #1: Yes

Reviewer #2: (No Response)

3. Has the statistical analysis been performed appropriately and rigorously? 

Reviewer #1: Yes

Reviewer #2: (No Response)

4. Have the authors made all data underlying the findings in their manuscript fully available?

Reviewer #1: Yes

Reviewer #2: (No Response)

5. Is the manuscript presented in an intelligible fashion and written in standard English?

Reviewer #1: Yes

Reviewer #2: (No Response)

6. Review Comments to the Author

Reviewer #1: The authors responded to all my comments and significantly improved the paper. I have no further comments.

Reviewer #2: Thank you to the authors for their interesting manuscript. It seems that there are several areas that need improvement. Here are some suggestions:

Abstract: The abstract should be concise and clearly state the purpose, significance, methods, results, and conclusions of the study. The abstract is still a bit long in the current version.

Introduction: The objectives in the introduction section should be clearly defined. It would be helpful to separate the objectives into distinct sections with clear boundaries and clear meanings. The current version is hard to understand.

Methods: The methods section is currently lengthy and lacks hierarchy. It would be beneficial to divide this section into more subsections to clearly separate and describe the different content. Each subsection should have a clear objective that aligns with the research purpose.

Results: The subheadings in the results section should be more aligned with the research objectives stated in the introduction. Currently, it is difficult to see a correlation between the subheadings and the objectives. Additionally, the results should be presented in a way that clearly explains the significance and contribution of the findings. This will help readers understand the true importance of the numbers, analysis results, and charts.

Discussion: The subheadings in the discussion section should also be clearer. There should be less overlap and repetition within the sections. The discussion should focus on outlining the key findings, explaining their biological or ecological mechanisms, speculating on possible reasons, and addressing the study's limitations and future research directions.

The language of the manuscript is still a bit difficult for me, who is not a native English speaker. These suggestions are only personal in nature and do not negate the value of the manuscript. I hope these suggestions help improve the clarity and structure of the manuscript.

7. PLOS authors have the option to publish the peer review history of their article (what does this mean?). If published, this will include your full peer review and any attached files.

Reviewer #1: **Yes: **Natalya Sergeevna Ivanova

Reviewer #2: No

---

## [Author Response · Author response to Decision Letter 1]

29 Feb 2024

Dear Reviewer,

Please accept our revised version of the manuscript Eurasian aspen (Populus tremula L.): Central Europe’s keystone species ‘hiding in plain sight’ PONE-D-23-20092R1. We edited the text, please see the file ‘Revised Manuscript with Track Changes.’ We concentrated our revisions on improving the Abstract and clarifying objectives (Introduction) so that they align well with section headings (Results, Discussion), as suggested by you. A detailed accounting of our revisions can be found in notes below, as well as in the Track Changes version of our manuscript included here.

We followed all your suggestions.

Abstract: The abstract should be concise and clearly state the purpose, significance, methods, results, and conclusions of the study. The abstract is still a bit long in the current version.

The abstract text body in the review 1 followed the criterion of the 300 words of the PlosOne guidelines. It was not “a bit long” however, we shorten the text to current 236 words along with some rewording. We think, it is concise and states “the purpose, significance, methods, results, and conclusions of the study” enough. 

Introduction: The objectives in the introduction section should be clearly defined. It would be helpful to separate the objectives into distinct sections with clear boundaries and clear meanings. The current version is hard to understand.

We deleted a potential confusion; one sentence before the specific goals. We also made the goals more concise by adjustment of (iii). We believe, goals are clearly defined. We are not sure how you want to “separate the objectives into distinct sections with clear boundaries and clear meanings”. What distinct section, where? It sounds like corrections in the next chapters as subsection.

Methods: The methods section is currently lengthy and lacks hierarchy. It would be beneficial to divide this section into more subsections to clearly separate and describe the different content. Each subsection should have a clear objective that aligns with the research purpose.

We created “the hierarchy” for the method section by dividing into subsections: Data sources, Aspen’s ecological characteristics and Analysis. We think the section is structured enough to be aligned with the research purpose. 

Results: The subheadings in the results section should be more aligned with the research objectives stated in the introduction. Currently, it is difficult to see a correlation between the subheadings and the objectives. Additionally, the results should be presented in a way that clearly explains the significance and contribution of the findings. This will help readers understand the true importance of the numbers, analysis results, and charts.

We aligned the Result subheadings with the Methods subheadings. There is a correlation between the subheadings and the objectives.

We are not sure, whatdo you mean by “the results should be presented in a way that clearly explains the significance and contribution of the findings. This will help readers understand the true importance of the numbers, analysis results, and charts.” The numbers are clearly linked to the analytical methods, charts and properly reported.

Discussion: The subheadings in the discussion section should also be clearer. There should be less overlap and repetition within the sections. The discussion should focus on outlining the key findings, explaining their biological or ecological mechanisms, speculating on possible reasons, and addressing the study's limitations and future research directions.

We have tried to improve the subheading titles to be more expressive. We think there is no overlap and repetition within the sections; the Aspen ecological potential section speaks, besides basic statistics of aspen, about the broad ecological potential clear from the constructed heatmaps. This potential was confirmed by the PCA and Random Forest analyses with enumeration of important environmental factors; the Aspen’s ecological niche section speaks about delineation of the niche of aspen as the ecological and growth optimum using the previous heatmap. There is no overlap with the previous section; the Aspen current distribution and pathways for improved habitat section discusses aspen management history, present state and reasons for its current distribution. Then, practical silvicultural practices were suggested for restoration, conservation, and adaptive management.

We think, there is no space for speculations in the Discussion since the analysis is based on the real data!

We believe this study is improved enough for publishing at the PLOS ONE.

Thank you very much for the improvement of the paper.

---

## [Decision Letter · Decision Letter 2]

12 Mar 2024

Eurasian aspen (Populus tremula L.): Central Europe’s keystone species ‘hiding in plain sight’

PONE-D-23-20092R2

Dear Dr. Kusbach,

We’re pleased to inform you that your manuscript has been judged scientifically suitable for publication and will be formally accepted for publication once it meets all outstanding technical requirements.

Kind regards,

Janusz J. Zwiazek

Academic Editor

PLOS ONE

Additional Editor Comments (optional):

Reviewers' comments:

Reviewer's Responses to Questions

**Comments to the Author**

1. If the authors have adequately addressed your comments raised in a previous round of review and you feel that this manuscript is now acceptable for publication, you may indicate that here to bypass the “Comments to the Author” section, enter your conflict of interest statement in the “Confidential to Editor” section, and submit your "Accept" recommendation.

Reviewer #1: All comments have been addressed

Reviewer #2: All comments have been addressed

2. Is the manuscript technically sound, and do the data support the conclusions?

Reviewer #1: Yes

Reviewer #2: Yes

3. Has the statistical analysis been performed appropriately and rigorously? 

Reviewer #1: Yes

Reviewer #2: Yes

4. Have the authors made all data underlying the findings in their manuscript fully available?

Reviewer #1: Yes

Reviewer #2: Yes

5. Is the manuscript presented in an intelligible fashion and written in standard English?

Reviewer #1: Yes

Reviewer #2: Yes

6. Review Comments to the Author

Reviewer #1: The authors responded to all the comments and significantly improved the paper. Now the paper meets all the strict requirements of a scientific journal and can be published.

Reviewer #2: The authors have carefully revised the manuscript and can be accepted in its current form. Good luck!

7. PLOS authors have the option to publish the peer review history of their article (what does this mean?). If published, this will include your full peer review and any attached files.

Reviewer #1: **Yes: **Natalya Ivanova

Reviewer #2: No

---

## [Editor Report · Acceptance letter]

15 Mar 2024

PONE-D-23-20092R2 

PLOS ONE

Dear Dr. Kusbach, 

I'm pleased to inform you that your manuscript has been deemed suitable for publication in PLOS ONE. Congratulations! Your manuscript is now being handed over to our production team.

Kind regards, 

on behalf of

Professor Janusz J. Zwiazek 

Academic Editor

PLOS ONE